# Paternal Inheritance of Bisphenol A Cardiotoxic Effects: The Implications of Sperm Epigenome

**DOI:** 10.3390/ijms22042125

**Published:** 2021-02-20

**Authors:** Marta Lombó, María Paz Herráez

**Affiliations:** 1Department of Animal Reproduction, INIA, Avenida Puerta de Hierro, 18, 28040 Madrid, Spain; mloma@unileon.es; 2Department of Molecular Biology, Faculty of Biology, University of León, Campus Vegazana s/n, 24071 León, Spain

**Keywords:** bisphenol A, paternal exposure, histone acetylation, sperm epigenome, heart development

## Abstract

Parental exposure to bisphenol A (BPA) has been linked to a greater incidence of congenital diseases. We have demonstrated that BPA induces in zebrafish males an increase in the acetylation of sperm histones that is transmitted to the blastomeres of the unexposed progeny. This work is aimed to determine whether histone hyperacetylation promoted by paternal exposure to BPA is the molecular mechanism underlying the cardiogenesis impairment in the descendants. Zebrafish males were exposed to 100 and 2000 µg/L BPA during early spermatogenesis and mated with non-exposed females. We analyzed in the progeny the expression of genes involved in cardiogenesis and the epigenetic profile. Once the histone hyperacetylation was confirmed, treatment with epigallocatechin gallate (EGCG), an inhibitor of histone acetyltransferases, was assayed on F1 embryos. Embryos from males exposed to 2000 µg/L BPA overexpressed the transcription factor *hand2* and the receptor *esr2b*, showing their own promoters—as well as that of *kat6a*—an enrichment in H3K9ac. In embryos treated with EGCG, both gene expression and histone acetylation (global and specific) returned to basal levels, and the phenotype was recovered. As shown by the results, the histone hyperacetylated landscape promoted by BPA in the sperm alters the chromatin structure of the progeny, leading to the overexpression of the histone acetyltransferase and genes involved in cardiogenesis.

## 1. Introduction

Congenital heart defects (CHDs) represent the main group of congenital anomalies and they are the leading cause of spontaneous abortions [1]. The worldwide incidence of CHDs is around 8 per 1000 live births, but this number keeps increasing and strongly depends on the country [2,3]. Even though 30% of all CHDs are related to gene mutations (mainly in those encoding cardiovascular transcription factors) [4,5], other circumstances such as maternal infections [6] or socioeconomic and environmental factors have been linked to these diseases as well [1]. In fact, several studies concluded that disruption of both male and female reproductive systems, triggered by occupational exposure to some chemicals during the periconceptional period, led to congenital malformations in the next generation [7,8,9,10]. In that regard, endocrine disruptors (EDCs) have been described as hazardous substances, being parental occupational exposure to these toxicants involved in CHDs [11]. Like other EDCs, bisphenol A (BPA) is able to bind to the receptors of many endogenous hormones: estrogens, androgens, thyroid hormone, and estrogen-related receptors [12]. Over the last two decades, a great many studies have confirmed the ubiquitous presence of this toxicant in the human body, since it is used for the manufacturing of a large number of plastic devices present in our daily life [13]. As a result of that, the adverse effects of BPA on neuroendocrine, reproductive, immune, and cardiovascular systems have been widely characterized [14]. Several signaling pathways that allow the cardiomyocyte differentiation and homeostasis are dependent on estrogenic responses [15]. In fact, it has been proved that embryonic exposure to concentrations ranging from 0.1 to 1 mg/L BPA induces estrogenic responses in zebrafish developing hearts, leading to cardiac edema [16]. Results from the same group have also demonstrated a plausible link between ERE (estrogen-responsive elements) activation and abnormalities in the formation of heart valves [17]. Moreover, in a previous study, we have shown that zebrafish embryos exposed to 2000 and 4000 µg/L BPA had greater levels of *esr2b* transcripts and a higher incidence of cardiac malformations [18].

Regarding the transmission of the effects to the next generation, it has been noticed that both maternal [19] and paternal [20] exposure to BPA lead to cardiogenesis impairment in rhesus monkeys and zebrafish, respectively. In fact, 20.65% of the progeny obtained from zebrafish males exposed to 2000 µg/L BPA suffer cardiac malformations, such as cardiac edema, incorrect looping, and showed disorganized heart walls [20]. In mammals, maternal exposure to BPA during pregnancy entails the direct contact of the embryos with the toxicant and its metabolites, which means a direct effect of BPA on heart formation [19]. Nevertheless, paternal inheritance requires a different pathway of transmission that is still unclear. Concerning male reproduction, BPA has been claimed to modify the information contained in the spermatozoa (DNA and remnant mRNAs), thus jeopardizing the paternal contribution to early embryo development [20,21,22]. The abundance of transcripts encoding insulin receptors, which are involved in cardiogenesis [23], dropped in the sperm of BPA-treated males as well as in the sperm and embryos of the next generations (F1 and F2) [20]. These transgenerational changes in the expression more are likely to rely on the transmission of an altered epigenetic profile than on a pure estrogenic effect of BPA.

Male germ cells undergo a unique epigenetic remodeling throughout the spermatogenesis, which consists on the acquisition of different covalent modifications in DNA or DNA-associated proteins that regulate gene expression without changing the genetic code [24,25]. Should chemical exposure alter the epigenetic reprogramming during maturation of the sperm cells, this could result in inheritance of incorrect marks that, when particularly affecting imprinted genes, might well lead to birth defects [26]. Although mammalian embryos undergo a complete epigenetic remodeling just after fertilization to avoid the inheritance of possible errors, there are some environmentally-induced epimutations that somehow escape this reprogramming and they are subsequently transmitted to following generations, even if they have not been in contact with the toxic environmental factor [27,28]. However, in zebrafish, there seems to be no epigenetic remodeling after fertilization [29,30]. Previous studies from our group confirmed that both DNA methylation and histone acetylation were increased in zebrafish testicular cells after BPA exposure [21]. Interestingly, the changes in histone acetylation of spermatozoa induced by BPA treatment were inherited by the progeny, which also showed increased levels of histone acetylation (H3K9ac and H3K27ac) [22]. Additionally, a similar increase in histone acetylation (H3K9ac and H4K12ac) was observed in zebrafish early embryos directly exposed to BPA, many of which developed heart malformations [18]. The maintenance of epigenetic marks is essential for proper heart formation [31,32]. In fact, hearts of mice suffering diabetes showed increased levels of H3 acetylation [33] and an increment of active histone marks, such as H3K27ac and H3K9ac, was reported after heart failure [34]. Moreover, mutations in genes encoding histone acetyltransferases have been recently associated with congenital heart defects [35]. These data point to the epigenetic effects, and specifically to histone acetylation, as an additional mechanism that may potentially render cardiac tissue more vulnerable to BPA.

Epigallocatechin gallate (EGCG), the major polyphenolic compound of green tea (7.1 g EGCG per 100 g green tea leaves), has been found to exhibit a large number of therapeutic properties for heart health: antioxidant, anti-inflammatory, anti-atherosclerosis, and anti-cardiac hypertrophy [36,37]. Since Kurutoniwa and colleagues first described the anti-estrogenic effects exerted by high doses of EGCG [38], more recent studies have focused on this potential to attenuate different pathologies, including breast cancer [39]. Furthermore, among all the catechins, EGCG displays the most powerful activity as histone acetyltransferase (HAT) inhibitor [40]. In fact, EGCG treatment during cardiac-specific differentiation of mesenchymal stem cells reduced the levels of H3K9 acetylation in the promoters of cardiac-related genes [41]. In previous work, we demonstrated that the co-treatment with EGCG counteracted the negative effects of BPA on early zebrafish embryos: the expression of estrogen receptors and the histone acetylation returned to basal levels, allowing proper embryonic development [18].

In this scenario, our hypothesis is that the cardiotoxic effects promoted by paternal BPA exposure are triggered by changes in the histone acetylation pattern. To confirm this hypothesis, zebrafish (*Danio rerio*) males were exposed to BPA during early spermatogenesis and F1 embryos were analyzed to evaluate the epigenetic profile and the expression of candidate genes related to cardiogenesis. Once the epigenetic changes were established, a rescue treatment with EGCG was assayed on the embryos.

## 2. Results

### 2.1. Transcriptomic and Epigenetic Changes Triggered by Paternal BPA Exposure

The expression of estrogen receptors as well as that of the transcription factors involved in heart formation was studied in the progeny obtained from males exposed to 100 and 2000 µg/L BPA. The results revealed that the expression of *esr2a* (estrogen receptor beta), *gper1* (G protein-coupled estrogen receptor), and *esrrga* (estrogen-related receptor gamma a) was not altered in F1 embryos at 3.3 hpf after paternal BPA exposure (Figure 1A). Nevertheless, *esr2b* (estrogen receptor beta) was overexpressed in 24 hpf embryos obtained from males treated with both doses of BPA (Figure 1B).

Concerning the expression of transcription factors, the results showed an increase in the expression of the basic helix-loop-helix (bHLH) transcription factor, *hand2*, in embryos at 24 hpf, when fathers were exposed to 2000 µg/L BPA; in contrast the expression of *gata5*, *gata4*, and *bmp4* remained steady in embryos obtained from BPA-exposed males (Figure 1C).

In order to check if changes in histone acetylation pattern lay behind the overexpression of *esr2b* and *hand2*, as well as in that of *kat6a* (reported in our previous work [22]), we analyzed the levels of H3K9ac in the promoters of these genes. Results revealed increased levels of this histone acetylation mark in the promoters of *kat6a*, *esr2b*, and *hand2* in 24 hpf embryos when exposing the fathers to the highest dose of BPA (Figure 1D). 

So as to confirm that the increase in this histone activating mark was actually the epigenetic change that led to the overexpression of *hand2* and *esr2b*, the DNA methylation of a CpG island in the exon 1 of *esr2b* and in the promoter of *hand2*—were CpG islands are located (UCSC Genome Browser)—was analyzed in 24-hpf-embryo obtained from control and BPA-exposed males and also when EGCG treatment was applied to F1 embryos. The results showed that the DNA methylation of these two regions did not change either in F1 embryos obtained from males exposed to BPA or in F1 embryos treated with EGCG (Figure 2).

### 2.2. Inhibition of BPA-Induced Histone Hyperacetylation with EGCG

Embryos obtained from control and from 2000 µg/L BPA-exposed males were treated with 50 µM EGCG, a powerful inhibitor of histone acetyltranferases, during the first 3 h of development. Histone acetylation was evaluated by whole mount immunostaining in F1 embryos from control and BPA-exposed males and results showed that, after treating embryos with EGCG, H3K9ac, and H3K27ac levels were similar to those of control groups (Figure 3A–D). In fact, the levels of H3K9ac in the promoters of *kat6a*, *esr2b*, and *hand2* were no longer significantly different between the embryos from control and BPA-exposed males when the treatment with EGCG was applied (Figure 3E). 

These data, together with the results of BPA paternal exposure previously published [22], have been normalized to their respective controls and are listed in Table 1, showing that the hyperacetylation observed in the progeny of treated males was no longer noticed when the embryos were treated with the catechin.

When it comes to gene expression profile, it was observed that EGCG treatment during the first 3 h of development successfully neutralized the overexpression of *esr2b* and *hand2* as well as *kat6a*, whose upregulation was also observed in 24 hpf embryos obtained from males exposed to 2000 µg/L BPA [22] (Figure 4). 

These data, together with the results of Figure 1 and those of BPA paternal exposure alone previously published [22], have been normalized to their respective controls and are listed in Table 1 and Table 2.

### 2.3. Mitigation of Cardiac Toxicity Induced by Paternal BPA Exposure

To confirm that the EGCG activity successfully counteracted the hazardous effects of paternal BPA exposure on embryonic development, especially on cardiogenesis, F1 embryos were monitored from 24 hpf to 7 dpf. Results revealed that, when EGCG treatment was applied to F1 embryos obtained from BPA-exposed males, the percentage of mortality remained the same as those of control groups (Figure 5A). Furthermore, these EGCG-treated embryos displayed a low percentage of cardiac malformations at 7 dpf (Figure 5B), similar to that of control embryos. These cardiac malformations included those described after BPA paternal exposure [20]: cardiac edema, problems in looping and ballooning and heart walls formation. 

These data, together with the results of BPA paternal exposure alone previously published [20,22], have been normalized to their respective controls and are listed in Table 1.

## 3. Discussion

The results derived from this study report, to our knowledge for the first time, how the paternally-induced epimutations caused by bisphenol A exposure display long-term effects on the development of the progeny, mainly on heart formation. Moreover, these findings confirmed the potential of EGCG to counteract the toxic effects triggered by paternal exposure BPA on cardiac health.

Cardiac malformations that result from failures in embryonic heart development represent the most prevalent cause of miscarriages and the main type of birth defects in humans [3]. In recent years, many studies have demonstrated that exposure to BPA during embryonic development impairs cardiogenesis [18,42,43,44]. Moreover, there is growing evidence that both maternal [19] and paternal [20] exposure to BPA also cause deleterious effects on the heart health of the progeny. 

Changes in gene expression are frequently associated with modifications in the epigenetic information. However, in our preceding study, in which the effects of direct embryonic BPA exposure were analyzed [18], we were not able to establish a link between the observed overexpression of the receptors and the transcription factors with an increase in histone acetylation on their specific promoters, pointing to a direct estrogenic effect as the most plausible mechanism for the observed gene dysregulation. In contrast, some remarkable differences indicate that epigenetic modifications could be a potential mechanism through which paternal exposure to BPA disrupts heart development in the progeny.

We have recently demonstrated that the epigenetic changes promoted by BPA exposure during early spermatogenesis in the zebrafish sperm, particularly the increased levels of histone acetylation, are inherited by F1 embryos [22]. Histone acetylation implies the neutralization of positive charges in lysine residues, reducing the interaction of histones and DNA, thus promoting the accessibility of chromatin to transcriptional machinery [45]. Therefore, we were able to establish a relationship between the histone hyperacetylation triggered by paternal BPA exposure and the observed upregulation of *esr2b* and *hand2* in the progeny. Indeed, ChIP-qPCR analysis revealed a higher enrichment of H3K9ac in the promoters of these genes. Additionally, this might well be related with the heart malformations caused by paternal BPA treatment we have previously described [20] since heart disease has already been linked to changes in histone acetylation [5,46]. This epigenetic modification is catalyzed by HATs, such as Kat6a that is in charge of H3 and H4 acetylation [47]. Not only was the expression of *kat6a* enhanced after embryonic BPA exposure [18], but it also increased in F1 embryos obtained from BPA-treated males [22]. This augmentation of *kat6a* expression in F1 embryos is also explained by the increment in H3K9ac of its own promoter, thus maintaining positive feedback of hyperacetylation. Besides, the absence of changes in DNA methylation in the promoters of these genes strengthens the fact that the alterations in the histone acetylation pattern bring about the embryonic heart failure observed in previous experiments [20]. To further confirm this hypothesis, a functional assay was developed. 

In a previous study, we determined that the effects of embryonic BPA exposure on estrogenic responses and histone acetylation were both neutralized by treating the embryos with 50 and 100 µM EGCG during the first 3 h of development [18]. EGCG, the most abundant catechin of green tea, has been shown to display a wide range of health-promoting effects, which make it emerge as an outstanding molecule to fight against cardiovascular diseases, as recently reviewed by Eng and colleagues and Cao and co-workers [36,37]. In this case, instead of a co-treatment of BPA with EGCG, we assayed the rescue of the paternal BPA effects on the next generation: 50 µM EGCG treatment was applied to embryos obtained from control males and males exposed to 2000 µg/L BPA from fertilization up to blastula stage (also the first three hours of development).

EGCG has anti-estrogenic effects [38] that have been attributed to its ability to compete with E_2_ for binding to both *ERα* and ERβ in vitro [48], but also to its capacity of blocking the transcription of *ERα* by recruiting the histone deacetylase HDAC1, among other co-repressor proteins, in the promoter of this gene [49]. In zebrafish, we confirmed that *esr2b* overexpression in 24 hpf embryos exposed to 4000 µg/L BPA was successfully counteracted after EGCG treatment [18]. Likewise, the present results confirmed that, when F1 embryos from BPA-exposed males are treated with 50 µM EGCG, the upregulation of *esr2b* returns to basal levels, being this reduction associated with a decrease in H3 acetylation. Owing to the ability of EGCG to inhibit 90% of HAT activity [40], the increased levels of H3K9ac and H3K27ac that were reported in F1 embryos obtained from males exposed to 2000 µg/L BPA [22] were mitigated after treatment of these embryos with 50 µM EGCG. The downstream alterations derived from the enhanced histone acetylation promoted by paternal BPA exposure were also alleviated: the levels of global histone acetylation were clearly reduced just after the treatment with EGCG had finished (at 3 hpf) and the overexpression of *hand2*, *esr2b*, and *kat6a* of F1 embryos obtained from BPA-treated males was reduced. In contrast, we only found a slight reduction in the enrichment of H3K9ac in the promoters of these genes at 24 hpf. This limitation in the potential of EGCG to decrease specific histone acetylation later on development is likely related to the short period of treatment we have used, which affects the early stages of cleavage until blastula is formed. Anyway, our results are similar to those observed in the experiment of Yin and colleagues [41]. In their experiment, the authors observed a decrease in H3 acetylation after having treated mouse mesenchymal stem cells with 120 μM EGCG during cardiac-specific differentiation, which led to a reduced expression of some genes crucial for this process: *Gata4*, *Nkx2.5*, and *Mef2c*. Besides, the treatment with EGCG we have used positively rescued the phenotype of F1 embryos: the high incidence of cardiac malformations, characteristic of the progeny from BPA-exposed males, returned to basal levels.

## 4. Materials and Methods

### 4.1. Ethics Statement

This work is included in a project from the Spanish Ministry of Economy and Competitiveness (Project AGL2014-53167-C3-3-R) specifically approved by the University of León Bioethical Committee as well as by the competent body of Junta de Castilla y León (project number: ULE009-2016). All the animals were manipulated in accordance with the Guidelines of the European Union Council (86/609/EU, modified by 2010/62/EU), following Spanish regulations (RD 1201/2005, abrogated by RD 53/2013) for the use of laboratory animals.

### 4.2. Zebrafish Maintenance and BPA Paternal Exposure

Eight-month-old zebrafish, AB strain (wildtype), were maintained in 1.5 L aquaria (ZebTEC, Tecniplast System) with a recirculating water system (pH 7.0–7.5, 30 mg/L Instant Ocean, at 27–29 °C, 14:10 light-dark cycle). Animals were fed twice a day with dry food (Special Diets Services^®^).

Adult males were exposed to ethanol and BPA in groups of four in glass tanks containing 1.5 L water. To carry out the experiments, males were maintained in water with the vehicle (ethanol 0.014% (*vol*/*vol*)) or 100 and 2000 μg/L BPA (0.44 and 8.8 μM, respectively) for two weeks, and the last week they were maintained in 1.5 L water. This schedule allows that, by the time of mating, mature spermatozoa have been exposed to BPA only during early spermatogenesis but it has not affected the cells during meiosis or spermiogenesis, which together last approximately one week in zebrafish [50]. Each treatment was repeated 8 times, using a total of 32 males per treatment. Water and BPA solutions were daily renewed throughout the experiment since BPA stability in this water has been confirmed for 24 h [21]. As far as the doses are concerned, we have used concentrations of BPA previously proved to alter embryonic development when exposing the fathers in this species [20]. The lowest one (100 µg/L) was the total allowed concentration in drinking water [51] and the highest one (2000 µg/L) represents a poisoning condition since it fits in the normal levels of BPA on waste landfill leachates [52] that promotes cardiac malformations in approximately 21% of the descendants [20]. The use of this dose is devoted to study the mechanisms of action (MoA) at high concentrations, fulfilling the requirements recommended by the EPA for the choice of dose selection to test an endocrine disruptive effect as it has been summarized by Marty and colleagues [53].

### 4.3. Embryo Collection and Experimental Design

After these 3 weeks, each of the four males per tank were mated with non-exposed females (one male with two females), giving a total of 32 mating groups per treatment, which allow obtaining the large number of embryos required to perform all the analytical techniques (three replicates per technique). Gene expression, histone acetylation marks, as well as acetylation and methylation of promoters were analyzed. In order to explore the potential reversion of paternal BPA effects using an inhibitor of histone acetylation, F1 embryos from control males and males exposed to 2000 μg/L BPA, were split up in two samples, being one of them treated with 50 µM EGCG from fertilization to blastula stage (3.3 h post fertilization, hpf) (Figure 6). This dose of EGCG has been already reported to counteract the molecular changes triggered by BPA embryonic exposure that lie behind cardiac malfromations [18]. Embryos were collected using a medium-mesh nylon net and they were transferred to a 60 mm-Petri dish containing embryo medium containing 0.038 mM CaCO_3_, 0.446 mM NaHCO_3_, 1.025 mM sea salt, and 0.005% (*vol*/*vol*) methylene blue at 28 °C in darkness until different probes were done. Embryo mortality was evaluated from 3.3 hpf each day until they reached 120 hpf. At 7 seven days post fertilization, phenotype of control and exposed larva were assessed under a stereomicroscope (Leica MZ16F). Transcriptomic and epigenetic analysis were performed at 3.3 and 24 hpf.

### 4.4. Gene Expression

Expression of candidate genes, related to estrogen signaling and to cardiogenesis, was analyzed in F1 embryos from control and BPA-exposed males, treated and non-treated with EGCG. RNA was extracted from a pool of 30 F1 embryos at 3.3 and 24 hpf, using Trizol reagent according to manufacturer’s protocol. The concentration of RNA samples was measured using the NanoDrop ND-1000 UV-Vis Spectrophotometer (Thermo Scientific), whereas the integrity was assessed by electrophoresis on agarose gel (data not shown).

The cDNA syntheses and RT-qPCR conditions as well as zebrafish primer sequences for the estrogen receptors (*esr2a*, *esr2b*, *esrrga*, *gper1*), the histone acetyltransferase (*kat6a*), the heart transcription factors (*hand2* and *gata5*) and the housekeeping genes (*18S rRNA* for 3.3 hpf-embryo and *actb2* for 24 hpf-embryo) were the same as those detailed by our group [18]. The RT-qPCR conditions and sequences for *gata4* and *bmp4* are described in Table 3. In zebrafish, other studies have reported that in vivo exposure to BPA alters the expression of *esr2a*, *esr2b*, *esrrga*, *gper1* [18,21,54], so we have analyzed their expression in embryos obtained from BPA exposed males with and without EGCG treatment. The expression of *kat6a* was evaluated since our previous results showed that paternal exposure to 2000 μg/L BPA led to an overexpression of this gene in F1 embryos at 24 hpf [22]. Regarding the times selected for gene expression, 3.3 and 24 hpf were selected because of the differential temporary expression of estrogen receptors (whose expression is maternally inherited until mid-blastula transition, at 3.3 hpf [55]) and of genes involved in heart tube formation at 24 hpf [56]. 

### 4.5. Whole Mount Immunostaining

For the whole mount immunostaining, 5 F1 control embryos and embryos treated with EGCG were collected at 3.3 hpf for each replicate, when changes in histone acetylation triggered by BPA where noticed [22], and fixed in 4% (*w*/*v*) PFA overnight at 4 °C. From this moment onwards, the protocol described by our group for the detection of H3K9ac (C5B11, Cell Signaling Technology^®^) and H3K27ac (ab4729, Abcam) was followed [18]. The potential neutralizing effects of EGCG treatment regarding histone acetylation were assessed in H3K9ac and H3K27ac, since their levels have significantly increased in the progeny after paternal BPA exposure [22]. Negative controls were performed incubating the embryos with blocking solution in the absence of primary antibodies. Images of blastomeres were taken by confocal microscopy (Zeiss LSM800) and histone acetylation was analyzed by measuring the intensity of the secondary antibody AlexaFluor^®^488 relative to the nuclear area stained with DAPI by using Image J software of, at least, 200 cells per replicate.

### 4.6. Histone Acetylation in Gene Promoters

Chromatin immunoprecipitation (ChIP) was performed in pools of 150 embryos at 24 hpf for each replicate and treatment following the protocol of Santos-Pereira and colleagues [57] to assess the enrichment of H3K9ac (ab12179, Abcam) in the promoters of *kat6a*, *esr2b*, and *hand2* at 24 hpf, when the expression of these genes was evaluated. An intergenic region of the genome lacking acetylation has been used to prove the absence of non-specific bindings. The same qPCR conditions and primer sequences as those described in our previous work [18] were used.

### 4.7. DNA Methylation in Gene Promoters

After DNA isolation from pools of 30 embryos at 24 hpf for each replicate and treatment, it was quantified using the NanoDrop ND-1000 UV-Vis Spectrophotometer (Thermo Scientific) and 600 ng of genomic DNA were treated with sodium bisulfite using EZ DNA Methylation™ Gold kit (Zymo Research), to selectively turn non-methylated cytosines into uracils. Then, total PCR was performed from 1 µL bisulfite-converted DNA and products were diluted 1:20 for the nested PCR. Programs for both total and nested PCR were: phase of 2 min at 94 °C followed by 40 cycles of 94 °C denaturation for 25 s, 58 °C annealing for 1 min and 72 °C extension for 2 min, and an extra step at 70 °C for 7 min. Then, 20 µL of products from nested PCR were mixed with annealing buffer and streptavidin beads. Once products were denaturated and annealing of pyrosequencing primers was achieved, final reactions were run with the PyroMark Q96 ID (Qiagen), which was also used for the analysis of CpG methylation that was calculated as the ratio of peak heights in the pyrogram. 

The sequence of the CpG islands near the transcription start site (TSS) were obtained from UCSC Genome Browser (https://genome.ucsc.edu/, accessed on 9 January 2021). The primers for total PCR, nested PCR, and pyrosequencing were designed using MethPrimer and they are listed in Table 3.

### 4.8. Statistical Analyses

Statistical analyses were performed with SPSS version 24.0 (IBM). For parametric data one-way ANOVA followed by DMS post hoc test was used, whereas for non-parametric data a Kruskal–Wallis test was applied following pairwise comparison and significance adjusted with Bonferroni correction. The type of test performed for each analysis as well as *p*-values obtained are indicated in Appendix A. All data in bars are represented as mean ± SEM and boxes represent median ± maximum and minimum. List of the different type of statistical analysis performed for each experiment, the post-hoc test used and the *p*-values obtained for each one are indicated in Appendix A.

## 5. Conclusions

In vivo zebrafish male exposure allowed us to find out the molecular mechanisms by which paternal exposure to BPA exerts long-term effects on F1 cardiogenesis: male exposure induces an increase in sperm histone acetylation that is inherited by the F1, altering the chromatin structure of crucial genes for heart development and that of the HAT in charge of maintaining the profile. Moreover, when F1 embryos obtained from BPA-exposed males were treated with EGCG, histone acetylation levels were restored, avoiding the overexpression of estrogen receptors and transcription factors and, therefore, reducing the probability of cardiac disease.

## Figures and Tables

**Figure 1 ijms-22-02125-f001:**
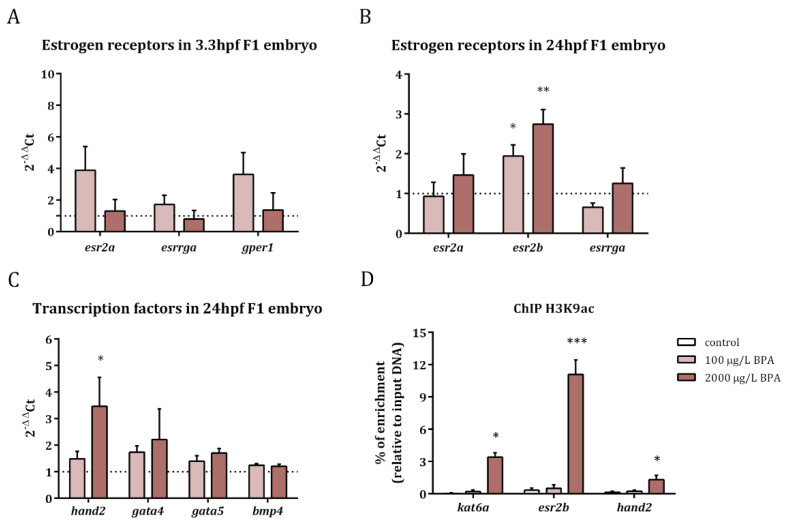
Relative expression of estrogen receptors in 3.3 hpf embryos (**A**) and 24 hpf embryos (**B**) and of transcription factors in 24 hpf embryos (**C**) from control and BPA-exposed males. Expression levels relative to *18SrRNA* and *actb2* were calculated using 2^−ΔΔ^Ct method of three replicates from pools of 30 embryos each one (*n* = 3). Asterisks indicate significant differences (* = *p* < 0.05) when comparing to control embryos (dashed line). Percentage of enrichment in H3K9ac of *kat6a*, *esr2b*, and *hand2* promoters in the progeny from control males and males exposed to BPA (**D**). Asterisks indicate significant differences (* = *p* < 0.05; ** = *p* < 0.01; *** = *p* < 0.001) when comparing to control group of three independent batches of 150-pooled 24 hpf embryos each one (*n* = 3).

**Figure 2 ijms-22-02125-f002:**
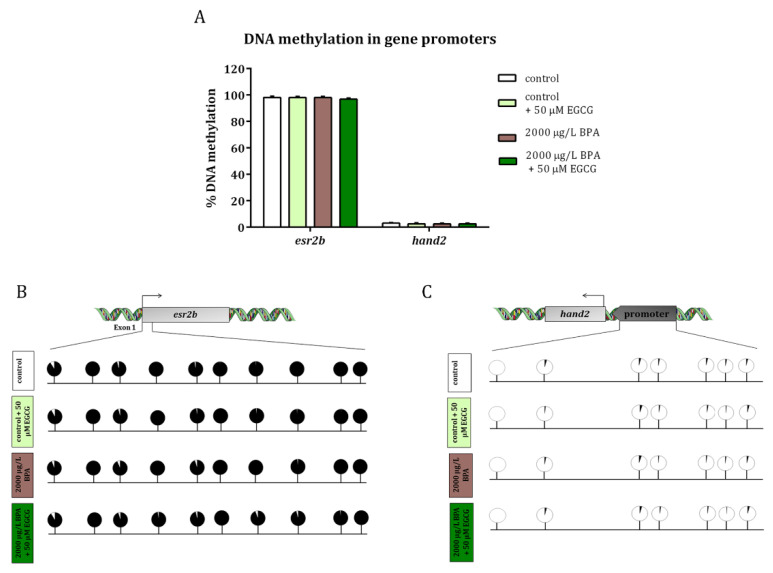
Analysis of DNA methylation in the exon 1 in *esr2b* and *hand2* promoter. (**A**) Bars show the percentage of methylation in 10 and 7 CpGs of each studied region, respectively, represented as mean ± SEM. (**B**,**C**) are images of the percentage of methylation in each individual CpG of three different replicates per treatment from pools of 30 embryos each one (*n* = 3). Black color represents the percentage of methylated CpGs, whereas white color indicates the percentage of non-methylated CpGs.

**Figure 3 ijms-22-02125-f003:**
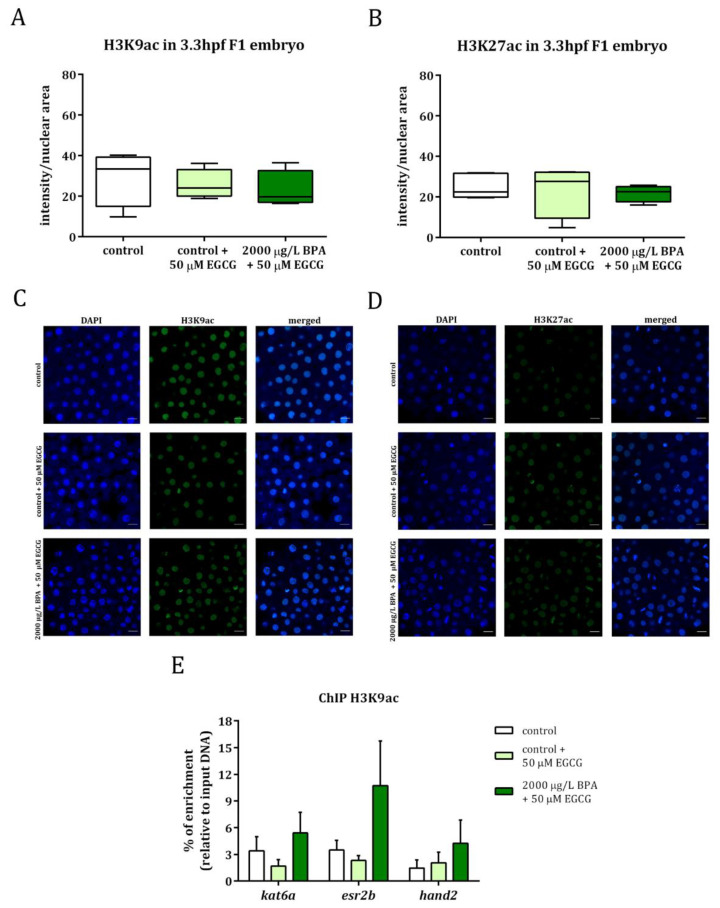
Analysis of histone acetylation in H3K9 (**A**) and H3K27 (**B**) in 3.3 hpf embryos from control and BPA-exposed males after 50 µM epigallocatechin gallate (EGCG) treatment. Boxes represent nuclear intensity of around 200 cells of 5 embryos per treatment (*n* = 5). Representative images of H3K9ac (**C**) and H3K27ac (**D**) in: control embryos, control embryos treated with 50 µm EGCG and embryos from 2000 µg/L BPA-exposed males treated with 50 µm EGCG. The different channels for DAPI and Alexa Fluor^®^ 488 appeared separated and merged; scale bar represents 20 µm. Percentage of enrichment in H3K9ac of *kat6a*, *esr2b*, and *hand2* promoters in the progeny from control males and males exposed to BPA after 50 µM EGCG treatment (**E**). Bars represent the mean ± SEM of three independent batches of 5, 24 hpf embryos each one (*n* = 3).

**Figure 4 ijms-22-02125-f004:**
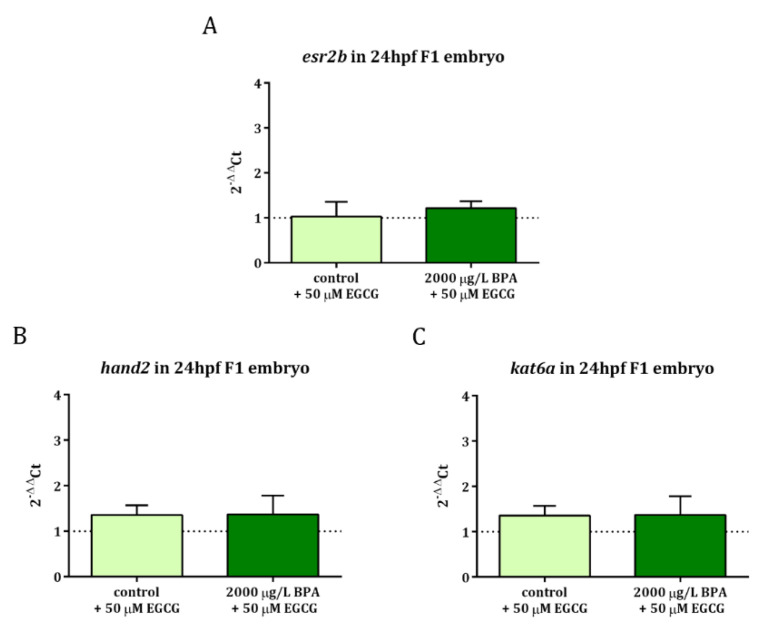
Analysis of gene expression: (**A**) *esr2b*, (**B**) *hand2*, and (**C**) *kat6a* in 24 hpf embryos from control and BPA-exposed males after 50 µM EGCG treatment. Expression levels relative to *actb2* were calculated using 2^−ΔΔ^Ct method of three replicates from a pool of 30 embryos each one (*n* = 3). Dashed line indicates the levels of gene expression in control embryos.

**Figure 5 ijms-22-02125-f005:**
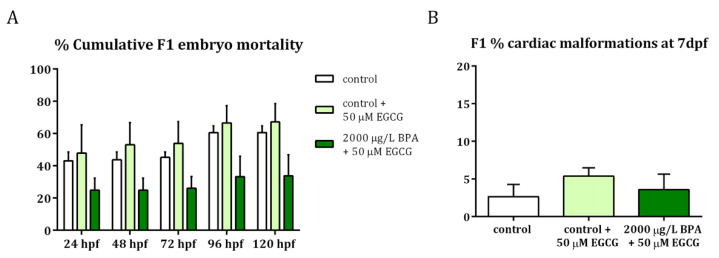
Evaluation of F1 embryos obtained from control and 2000 µg/L BPA-exposed males after EGCG treatment. (**A**) Percentage of F1 embryo mortality throughout the development. (**B**) Percentage of cardiac malformations in F1 larvae at 7 dpf. Bars represent mean ± SEM of 3 different experiments (*n* = 3).

**Figure 6 ijms-22-02125-f006:**
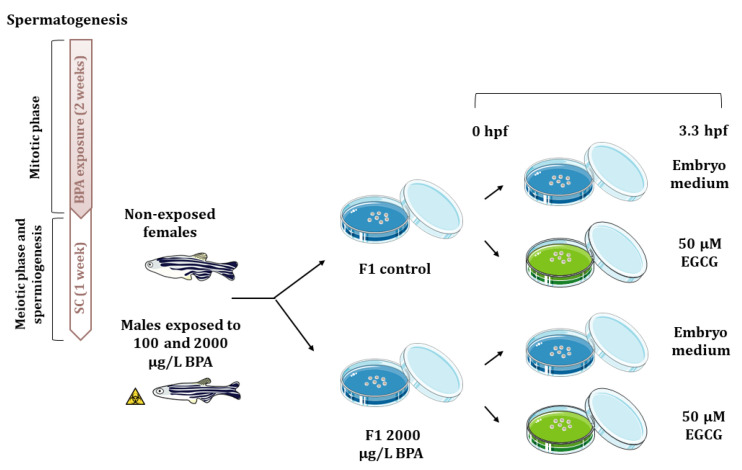
Schematic representation of the experimental design. Adult males were exposed to 0.014% ethanol and 100 or 2000 µg/L BPA for two weeks and after that, they were maintained one week under standard conditions. Then, males were mated with non-treated females and the F1 progeny from control males and males exposed to 2000 µg/LBPA was treated with 50 µM EGCG from fertilization up to 3.3 hpf.

**Table 1 ijms-22-02125-t001:** Data obtained in this work from F1 embryos treated with 50 μM EGCG and published data from the progeny of males exposed to 2000 μg/L BPA were normalized to their respective controls. Data are expressed as mean ± standard deviation.

	Normalized Published Data		Normalized Original Data	
	F1 Embryos from Males Exposed to 2000 μg/L BPA	Reference	F1 Embryos from Control Unexposed Males Males, Treated with 50 μM EGCG	F1 Embryos from Males Exposed to 2000 μg/L BPA, Treated with 50 μM EGCG
**H3K9ac**	5.24 ± 1.71	[22]	0.88 ± 0.25	0.79 ± 0.31
**H3K27ac**	2.79 ± 1.04	[22]	0.92 ± 0.51	0.87 ± 0.16
***kat6a* expression (2^−ΔCt^)**	4.29 ± 5.32	[22]	1.14 ± 0.24	0.64 ± 0.31
**Mortality at 24 hpf**	2.27% ± 0.25	[22]	1.11% ± 0.7	0.58% ± 0.3
**Mortality at 48 hpf**	2.18% ± 0.24	[22]	1.21% ± 0.54	0.57% ± 0.3
**Mortality at 72 hpf**	1.97% ± 0.21	[22]	1.19% ± 0.52	0.58% ± 0.28
**Mortality at 96 hpf**	1.56% ± 0.14	[22]	0.38% ± 0.54	0.006% ± 0.009
**Mortality at 120 hpf**	1.59% ± 0.11	[22]	1.11% ± 0.32	0.55% ± 0.37
**Malformations At 7 dpf**	5.61 ± 0.49	[20]	2.01 ± 0.72	1.35 ± 1.34

**Table 2 ijms-22-02125-t002:** Data regarding the expression of *esr2b* and *hand2* (expressed as 2^−ΔCt^) were normalized to their respective control and are expressed as mean ± standard deviation.

	F1 Embryos from Males Exposed to 2000 μg/L BPA	F1 Embryos from Control Unexposed Males Males, Treated with 50 μM EGCG	F1 Embryos from Males Exposed to 2000 μg/L BPA, Treated with 50 μM EGCG
***esr2b* expression (2^−ΔCt^)**	2.71 ± 0.56	0.77 ± 0.27	1.18 ± 0.85
***hand2* expression (2^−ΔCt^)**	2.15 ± 1.38	1.14 ± 0.24	0.64 ± 0.31

**Table 3 ijms-22-02125-t003:** Primers designed for RT-PCR, gene promoter methylation, and pyrosequencing. Sequences start from 5′ to 3′. Product size is indicated as number of base pairs (bp) and annealing temperature is indicated in Celsius scale (°C). Annealing temperature for total and nested PCR was 58 °C.

**Primers Used for qPCR**					
**Gene**	**Primers**	**Product Size (bp)**	**Temperature (°C)**	**Accesion Number**	**Efficiency (%)**
*gata4*	F: CCGCTCGTGGAGCAATAATC R: CTGGATCATCGGAGTCACCC	154	64	DQ886664.1	92
*bmp4*	F: GCGCTGGACCCAAGAAAAAC R: TTGCCGTCATGTCCGAATGT	177	64	NM_131342.2	99
**Primers Used for Gene Promoter Methylation**					
**Gene**	**Forward Primer**	**Reverse Primer**		**Product Size**	**Use**
*esr2b*	TAGGTTAGGGTTTTTTTTGT	AAACTAAATTATTCTCACCTACTC		439	Total PCR
*hand2*	TATTTTTTGAGTTGTTTGGG	CCCTTCACCAAAAATTTTAA		549	Total PCR
*esr2b*	GAGGTTTGTTAGGATTATTTTTT	ATATATCTTAACCTCCTCCC		233	Nested PCR
*hand2*	TTAAAAGTAGTTAATTTATTGGT	ACTAATCCTTATACTACATTC		295	Nested PCR
**Primers Used for Pyrosequencing**					
**Sequences for *esr2b***		**Product Size**	**Sequences for *hand2***		**Product Size**
GTTTGTTAGGATTATTTTTT		28	TTATTTATTTAAAAAAAAAA		36
TTTAATTACGGAGTTTTA		18	GGTTTTTTTTTTTTTTTAGTGTGTG		26
TTGTAGTGTTCGGTTTTT		33	TAAATTAGTTTAAGTATATT		15
GTCGTTTTATTTTTTGTAT		34			

## Data Availability

The data presented in this study are available on request from the corresponding author.

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
