# Peer review of "Paternal Inheritance of Bisphenol A Cardiotoxic Effects: The Implications of Sperm Epigenome"

_ijms, 2021, doi:10.3390/ijms22042125_

Round 1

Reviewer 1 Report

Lombó and Herráez have provided an interesting report addressing the epigenetic alterations induced in zebrafish males due to bisphenol A exposure and the potential of transmission to the next generation through the paternal line. According to the provided results, paternal exposure to this endocrine disruptor seems to alter the levels of specific estrogen receptors, transcription factors and histone acetylation in F1 embryos.

In addition, authors have also explored the chances to overcome the toxic effects on the F1 by the treatment with EGCG during embryo development. However, one of the major concerns of this manuscript is that no comparison between treated and non-treated embryos is shown in any of the figures. This fact does not allow the evaluation of the actual effect of EGCG in the rescue of the molecular alterations that may induce cardiac malformations. Authors should include data without EGCG treatment in figures 3, 4 and 5.

Other major points in the experimental design should be also clarified before to consider this study suitable for publication, as follows:

  • Authors should explain the rationale behind the specific selection of histone modifications H3K9ac and H3K27ac to evaluate alterations in histone acetylation. Please, justify why the analysis of only these two modifications in Histone H3 might be used to set conclusions about the global state of histone acetylation in the embryos, as the authors did in the manuscript (see for instance Lines 256 and 261). Authors should clarify why the acetylation in other histone variants or even the evaluation of total acetylation was not addressed in the study.
  • In figure 1, significant changes in the expression of esr2b and hand2 are shown in 24 hpf F1 embryos. However, in subsequent analyses, the authors evaluated the effects of EGCG treatment not only on these two genes, but also on kat6a (Figure 4). Please, provide a rationale for the evaluation of the selected genes.

Other points to be addressed:

  • In the methods section, it is not completely clear how many males and embryos were used in each assay. Please, indicate clearly the amount of subjects included in each assay and triplicate.
  • Authors should provide statistical data after applying a post-hoc correction test, in order to better evaluate the robustness of the statistical results.
  • Figure 5: lower embryo mortality is observed in those embryos treated with 2000 ug/L BPA + 50 uM EGCG, in comparison with control groups. How can this be explained? As indicated before, authors should also include data about the treatment of F1 embryos only with BPA.
  • Lines 264-267: Did the authors consider evaluating whether the treatment of BPA exposed males might prevent the effects on the progeny?
  • Lines 115-117: delete this part
  • Line 123: indicate gene esrrga in italics

Author Response

First of all, we would like to thank you for your time and considerations. We have taken your suggestions and corrections into account and we hope that the following version of our manuscript will meet your requirements.

Reviewer 1:

Lombó and Herráez have provided an interesting report addressing the epigenetic alterations induced in zebrafish males due to bisphenol A exposure and the potential of transmission to the next generation through the paternal line. According to the provided results, paternal exposure to this endocrine disruptor seems to alter the levels of specific estrogen receptors, transcription factors and histone acetylation in F1 embryos.

In addition, authors have also explored the chances to overcome the toxic effects on the F1 by the treatment with EGCG during embryo development. However, one of the major concerns of this manuscript is that no comparison between treated and non-treated embryos is shown in any of the figures. This fact does not allow the evaluation of the actual effect of EGCG in the rescue of the molecular alterations that may induce cardiac malformations. Authors should include data without EGCG treatment in figures 3, 4 and 5.

The effects of the paternal BPA exposure on the progeny had been already evaluated and published [1,2]. In order to allow a better understanding of the recovery achieved with the EGCG treatment, we have normalized the relevant data respect to their controls and we have displayed them in a new table, allowing an easy comparison among the values.  

Other major points in the experimental design should be also clarified before to consider this study suitable for publication, as follows:

  • Authors should explain the rationale behind the specific selection of histone modifications H3K9ac and H3K27ac to evaluate alterations in histone acetylation. Please, justify why the analysis of only these two modifications in Histone H3 might be used to set conclusions about the global state of histone acetylation in the embryos, as the authors did in the manuscript (see for instance Lines 256 and 261). Authors should clarify why the acetylation in other histone variants or even the evaluation of total acetylation was not addressed in the study.

The rationale for the choice has been included in the M&M (section 4.5): among the four acetylation marks we have analyzed after paternal BPA exposure (H3K9ac, H3K27ac, H3K14ac, H4K12ac) we only observed changes in H3K9ac and H3K27ac of sperm cells, which were transmitted to the F1 embryo. Therefore, we have assessed the recovery of normal values in these two histone marks after EGCG treatment.

  • In figure 1, significant changes in the expression of esr2b and hand2 are shown in 24 hpf F1 embryos. However, in subsequent analyses, the authors evaluated the effects of EGCG treatment not only on these two genes, but also on kat6a (Figure 4). Please, provide a rationale for the evaluation of the selected genes.

Like other EDCs, BPA is able to interfere with the signaling pathways of a great variety of hydrophobic molecules, including estrogens, androgens and thyroid hormones. Out of the nine mechanisms of action previously described for EDCs, BPA mainly exerts its effects by binding to hormone receptors or by modifying their expression [3].

It is noteworthy that estrogen receptors are ERα and ERβ in mammals and Esr1 and Esr2 in zebrafish. BPA can act either as agonist or antagonist of estrogens, this activity depending on the ER subtype and the tissue involved [4]. Moreover, BPA exhibits several mechanisms of action to interfere with GPER signaling pathway. Although the affinity of estradiol for GPER is 10-fold lower than for ERα [5], BPA displays an outstanding capacity to join GPER [6]. It has also been proved that both in vivo male exposure and in vitro testicular exposure to BPA lead to an increase in gper expression and Gper protein level in zebrafish [7]. Regarding zebrafish estrogen-related receptors, the high affinity of ERRγ for BPA has been reported to impair otolith formation, this process being reversed by BPA absence or/and morpholino knockdown of ERRγ [8].

Taking all these data into account, we selected the mentioned genes since BPA was likely to interfere with their expression, as it has been indicated in M&M section 4.4 . Nonetheless, out of the four genes studied, paternal exposure to BPA only led to an overexpression of esr2b.

On the other hand, the expression of kat6a was also evaluated since our previous results showed that paternal exposure to 2000 μg/L BPA led to an overexpression of this gene in F1 embryos at 24 hpf [1], as it has been stated in M&M section  4.4.

Other points to be addressed:

  • In the methods section, it is not completely clear how many males and embryos were used in each assay. Please, indicate clearly the amount of subjects included in each assay and triplicate.

The number of embryos used in each study has been mentioned in the M&M section as well as in the Figure legends.

  • Authors should provide statistical data after applying a post-hoc correction test, in order to better evaluate the robustness of the statistical results.

A new table has been included as supplementary material to show the statistical post-hoc and the p-values.

  • Figure 5: lower embryo mortality is observed in those embryos treated with 2000 ug/L BPA + 50 uM EGCG, in comparison with control groups. How can this be explained? As indicated before, authors should also include data about the treatment of F1 embryos only with BPA.

Embryos were never treated with BPA, only parents were treated. It is noteworthy that the mortality of control embryos and those from BPA-exposed males treated with EGCG were not statistically significant. The data about embryos from BPA-exposed males which were not rescued with EGCG, have been included in Table 1.

  • Lines 264-267: Did the authors consider evaluating whether the treatment of BPA exposed males might prevent the effects on the progeny?

Yes, in fact we have published a paper showing that the effects of direct embryo exposure to BPA were also rescued by a co-treatment with EGCG. So it is likely that co-treatment of males with BPA and EGCG, could reduce the effects of BPA on the sperm and, consequently, on the progeny. However, our aim was to confirm our hypothesis that the effects on the progeny were promoted by the inheritance of an enhanced histone hyperacetilation activity, so we decided that the inhibition should be done on the embryos. Moreover, we have also born in mind that EGCG could have different effects during spermatogenesis, including changes in the transcriptomic profile of the spermatozoa, which could make more difficult the analysis of the mechanism of transmission to the next generation.

  • Lines 115-117: delete this part

It was a mistake and it has been removed.

  • Line 123: indicate gene esrrga in italics

It has been corrected.

Reviewer 2 Report

General comments: This research addresses important questions about the toxicity of BPA on cardiac parameters in males. The research is well done, tells an interesting story, and contributes to the field of toxicology in a profound manner. However, there are some questions regarding a better understand of the work.

  1. The authors note that their experimental replicated were 3. However, the number of fishes from where the samples are collected is not mentioned. Please list sample size clearly in figure legends.
  2. Because bisphenol A is rapidly metabolized in the body, some in vitro studies use BPA metabolites instead of the parent compound. Would it be more likely that cardiac cells in the body would come into contact with BPA metabolites rather than the parent compounds? If so, can the authors justify why they used a parent compound instead of its metabolite?
  3. Do authors conduct any dose/time-dependent experiment? How exposure limits are relevant to human-relevant doses?
  4. As positive control is not considered, it is very difficult to say whether the cardiotoxic effects of BPA due to the endocrine disruption.
  5. The procedure for embryo collection should be described more elaborately for a better understanding of the method.
  6. The manuscript has grammatical and typographical errors throughout and should be edited.
  7. Need to be deleted “This section may be divided by subheadings. It should provide a concise and precise description of the experimental results, their interpretation, as well as the experimental conclusions that can be drawn.”

Author Response

Reviewer 2:

General comments: This research addresses important questions about the toxicity of BPA on cardiac parameters in males. The research is well done, tells an interesting story, and contributes to the field of toxicology in a profound manner. However, there are some questions regarding a better understand of the work.

  1. The authors note that their experimental replicated were 3. However, the number of fishes from where the samples are collected is not mentioned. Please list sample size clearly in figure legends.

The simple size has been specified in Figure legends and in the M&M section

  1. Because bisphenol A is rapidly metabolized in the body, some in vitro studies use BPA metabolites instead of the parent compound. Would it be more likely that cardiac cells in the body would come into contact with BPA metabolites rather than the parent compounds? If so, can the authors justify why they used a parent compound instead of its metabolite?

We have not analyzed the direct effect of BPA on cardiac cells. Embryo hearts are suffering the effects of the paternal treatment with BPA. This means that we have treated the males with BPA, fish have metabolized the compound and BPA or its metabolites have caused an effect on the spermatozoa, which in turn has promoted an effect on the embryo gene expression leading to cardiac malformations. The experimental procedure reproduces the exposure of the specimens to the toxic in water.

  1. Do authors conduct any dose/time-dependent experiment? How exposure limits are relevant to human-relevant doses?

Not, we haven’t. This dose/time-dependent experiments are more proper of in vitro or embryonic exposure in which the windows of exposure are shorter and the direct effects of the toxicants can be observed in both cells and tissues. Nonetheless, in a previous work we have conducted an experiment in which we have use two different windows of exposure: the early spermatogenesis, (comprising the mitotic period), and the whole spermatogenesis (in which also meiosis and spermiogenesis were affected) [1]. Taking into account the results we have observed, we decided to perform for the current experiment the first type of exposure, only affecting the early phases of spermatogenesis, since it resulted much less drastic for embryo survival.

As was reviewed by Gao and Wang [9], research about BPA effects could be categorized in two groups: the first one includes the “low dose” BPA experiments, which try to figure out the effects of environmentally relevant doses; whereas the studies belonging to the second group apply supra-physiological doses of BPA to identify the toxicological actions involved in specific poisoning conditions or in occupational exposure. This work is not a pure toxicological study intended to give information about the dose-response effects. Here, we have used concentrations of BPA previously proved to alter embryonic development when exposing the fathers [2]. The lowest one (100 µg/L) is environmentally relevant, as it is explained in materials and methods, and the highest one (2000 µg/L) represents the poisoning conditions since it fits in the normal levels of BPA on waste landfill leachates [10]. The use of this dose is devoted to study the mechanisms of action (MoA) at high concentrations, fulfilling the requirements recommended by the EPA for the choice of dose selection for screening/testing an endocrine disruptor. As it has been summarized by Marty et al [11], the EPA recommends setting the highest dose level at the maximum tolerated dose or concentration (MTD or MTC). For aquatic testing the MTC is calculated as a function of median lethal concentrations. The current MTC guidance in the test guideline is to set the highest tested concentration at one third of the fish acute 96-h LC50 (the dose inducing 50% mortality at 96 hpf). In zebrafish, this dose has been established in 12 mg/L [12], so we are still below half the highest recommended dose, that would be 4 mg/L.

In no case we would dare to extrapolate our results to humans, but to other aquatic organisms. We are aware that humans are exposed to much lower doses of BPA, but during a lifetime. Here, we have exposed zebrafish males during two weeks. However, the doses we have used are indeed much higher than that reported for humans but still, they have been used in several works in this species [13–15].

  1. As positive control is not considered, it is very difficult to say whether the cardiotoxic effects of BPA due to the endocrine disruption.

BPA has been described as an endocrine disruptor but, additionally, it has genotoxic and epigenotoxic effects. Considering our experimental procedure, endocrine, genotoxic and epigenotoxic effects are expected in the testicles and germinal cells of the exposed males. It seems unlikely to us that the effects on embryo development (transmitted by the spermatozoa during fertilization) could have been caused by endocrine disruption because there is not a direct contact between the embryos and the BPA. Regarding to the positive control, data have been included in Table 1.

  1. The procedure for embryo collection should be described more elaborately for a better understanding of the method.

More information has been added to this section.

  1. The manuscript has grammatical and typographical errors throughout and should be edited.

We have corrected the errors detected throughout the manuscript.

  1. Need to be deleted “This section may be divided by subheadings. It should provide a concise and precise description of the experimental results, their interpretation, as well as the experimental conclusions that can be drawn.”

It was a mistake and it has been removed.

Round 2

Reviewer 1 Report

The authors have addressed all the queries and provided an improved version of the manuscript. The effort of the authors is acknowledged. This reviewer has no more comments to add to this recommendation.

Reviewer 2 Report

The authors have dealt with my suggestions satisfactorily.